# Accident Anticipation via Temporal Occurrence Prediction

**Tianhao Zhao**[1,2]    **Yiyang Zou**[1]    **Zihao Mao**[1]    **Peilun Xiao**[3]    **Yulin Huang**[3]

**Hongda Yang**[3]    **Yuxuan Li**[3]    **Qun Li**[3]    **Guobin Wu**[3]    **Yutian Lin**[1]*

[1]School of Computer Science, Wuhan University    [2]Zhongguancun Academy, Beijing, China
[3]Didi Chuxing
{happytianhao, yutian.lin}@whu.edu.cn

## Abstract

Accident anticipation aims to predict potential collisions in an online manner, enabling timely alerts to enhance road safety. Existing methods typically predict frame-level risk scores as indicators of hazard. However, these approaches rely on ambiguous binary supervision—labeling all frames in accident videos as positive—despite the fact that risk varies continuously over time, leading to unreliable learning and false alarms. To address this, we propose a novel paradigm that shifts the prediction target from current-frame risk scoring to directly estimating accident scores at multiple future time steps (e.g., 0.1s–2.0s ahead), leveraging precisely annotated accident timestamps as supervision. Our method employs a snippet-level encoder to jointly model spatial and temporal dynamics, and a Transformer-based temporal decoder that predicts accident scores for all future horizons simultaneously using dedicated temporal queries. Furthermore, we introduce a refined evaluation protocol that reports Time-to-Accident (TTA) and recall—evaluated at multiple pre-accident intervals (0.5s, 1.0s, and 1.5s)—only when the false alarm rate (FAR) remains within an acceptable range, ensuring practical relevance. Experiments show that our method achieves superior performance in both recall and TTA under realistic FAR constraints. Project page: https://happytianhao.github.io/TOP/

## 1   Introduction

Driving accidents pose a significant threat to public safety, resulting in substantial human casualties and economic losses. This issue often arises when drivers, due to fatigue or distraction, fail to notice potential hazards in their surroundings, ultimately resulting in accidents. Recently, the task of accident anticipation [1, 2] has been widely studied to analyze the risk in driving scenarios captured by dashcams in an online manner, assessing the likelihood of an impending accident with a risk score, as shown in Figure 1 (a). If the risk score exceeds a preset threshold, the system can promptly alert the driver to take evasive action, thereby reducing the chances of an accident or mitigating its severity.

Previous works [3, 4, 5, 6, 7, 8, 9] train models to predict frame-wise risk scores using binary supervision: all frames from accident videos are labeled as 1, and all frames from safe videos as 0. To reflect the intuition that risk increases near the crash, these methods typically assign exponentially decaying loss weights to frames based on their temporal distance to the accident—earlier frames receive lower weights, while those near the crash receive higher ones. However, this approach treats

---

*Corresponding author

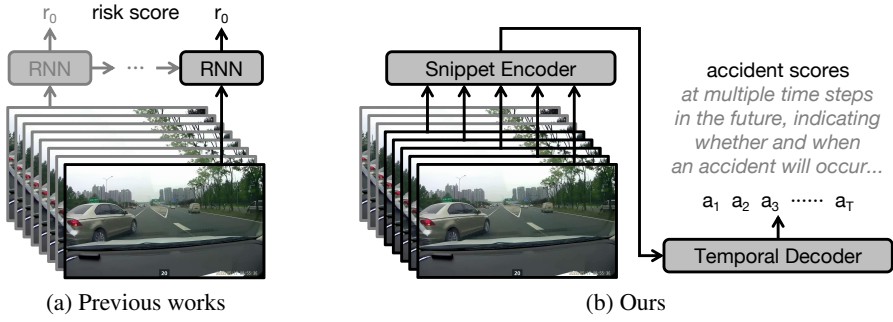

(a) Previous works                 (b) Ours

Figure 1: Comparison of accident anticipation paradigms. (a) Previous works predict a single risk score for the current frame, which is ambiguous and hard to supervise accurately. (b) Our method predicts a sequence of accident scores at multiple future time steps (e.g., 0.1s, 0.2s, . . . , 2.0s ahead), where each score indicates the likelihood of an accident occurring exactly at that future time.

risk as a static binary signal, ignoring its dynamic and continuously varying nature. In reality, risk levels differ significantly across frames even within the same accident sequence, and assigning a uniform label of 1 fails to capture these temporal nuances. Such imprecise supervision misguides learning and often leads to unreliable predictions or false alarms.

We observe that, unlike ambiguous risk labels, the timestamp of actual accident occurrence can be precisely annotated in real-world driving videos. To leverage this reliable supervision, we shift the prediction target from per-frame risk scores to directly forecasting future accident timing. Specifically, as shown in Figure 1 (b), our model outputs a sequence of accident scores at multiple future time steps $\{a_1, a_2, \ldots, a_T\}$ (e.g., 0.1s, 0.2s, . . . , 2.0s ahead), where each score $a_t$ indicates the model's confidence that an accident occurs exactly at that time. During training, only the score at the ground-truth accident time is labeled as 1; all others are 0. At inference, an alert is triggered if any score exceeds a preset threshold, indicating an impending collision. This formulation offers two key advantages: (1) it uses precise temporal annotations for more stable and accurate training, and (2) by modeling *when* an accident may occur—rather than just *whether* the scene is "risky"—it yields more interpretable and actionable predictions.

To implement this paradigm online, we adopt an encoder–decoder architecture that processes current and past frames to predict accident scores for multiple future time steps (e.g., 0.1s, 0.2s, . . . , 2.0s ahead), as shown in Figure 1 (b). Unlike previous works [3, 4, 5, 6, 7, 9], which typically use frame-level encoders with RNNs to model temporal dynamics, our method employs a snippet-level encoder that jointly captures spatial and temporal information across short clips of consecutive frames, enabling a comprehensive understanding of object positions, speeds, and motion trajectories. Furthermore, we design a Transformer-based temporal decoder that simultaneously predicts accident scores for all future time steps using distinct learnable temporal queries—each corresponding to a specific future horizon—to explicitly model accident likelihood at each time offset, supporting accurate and efficient frame-by-frame online prediction.

To evaluate the effectiveness of accident anticipation methods, previous works [3, 4, 5, 6, 7, 9] primarily use AP (Average Precision), AUC (Area Under the ROC Curve), and TTA (Time-to-Accident). However, we observe that in real-world applications, an excessively high false alarm rate (FAR)—e.g., exceeding 1 false alarm per minute—causes disruptive alerts that are unacceptable; under such conditions, high recall may stem from indiscriminate alarming rather than genuine prediction, rendering recall and TTA misleading. To address this, we propose a novel evaluation protocol that reports mean recall and TTA only when FAR is within an acceptable range. Furthermore, we evaluate recall at different pre-accident intervals (0.5s, 1.0s, and 1.5s before crashes) for a more granular assessment of anticipative capability. Finally, we identify limitations in existing TTA calculations that yield inflated values and propose a more reasonable approach, as detailed in Section 4.

Our contributions can be summarized as follows:

- We propose a novel accident anticipation paradigm that shifts from predicting ambiguous per-frame risk scores to directly estimating accident scores at multiple future time steps (e.g., 0.1s, 0.2s, . . . , 2.0s ahead), leveraging precise accident timestamps as supervision for more accurate and interpretable predictions.

- We design an effective encoder–decoder architecture featuring a snippet-level encoder to jointly capture spatial and temporal features from driving scenarios, and a Transformer-based temporal decoder that predicts accident scores for all future time steps simultaneously using dedicated temporal queries, enabling online frame-by-frame anticipation.

- We introduce a refined evaluation protocol that computes recall and Time-to-Accident (TTA) only when the false alarm rate (FAR) is within an acceptable range, evaluates recall at multiple pre-accident intervals (0.5s, 1.0s, 1.5s), and adopts an improved TTA calculation method that avoids inflated values, offering a more reliable and practical assessment.

## 2 Related Work

### 2.1 Temporal Modeling and Attention-Based Approaches

Early works on accident anticipation primarily rely on recurrent architectures to model temporal dynamics in dashcam videos. DSA [3] introduced the first large-scale dataset (DAD) and combined object-level and frame-level features with an RNN to predict per-frame risk scores, using an exponentially decaying loss that emphasizes frames closer to the accident. Subsequent methods enhanced temporal modeling through attention mechanisms. ACRA [2] proposed a soft-attention RNN to capture spatial and appearance interactions between the event-triggering agent and its surroundings. AdaLEA [10] improved early anticipation via an adaptive loss weighting strategy, while DSTA [11] introduced dynamic spatial-temporal attention to focus on relevant regions over time.

More recently, transformer-based architectures have emerged. AAT-DA [12] integrates driver attention into a transformer framework to jointly model spatial and temporal cues. LATTE [13] further advances temporal modeling by combining multiscale spatial features with memory-based attention and auxiliary self-attention for long-range dependency capture. Meanwhile, XAI [14] employs a GRU-based network to learn spatio-temporal relations, and RARE [15] achieves efficiency by leveraging intermediate features from a single pre-trained detector. THAT-Net [16] enhances motion understanding by fusing optical flow with spatial-temporal filtering to suppress distracting motions.

### 2.2 Graph-Based and Relational Reasoning Methods

To explicitly model interactions among traffic participants, several works adopt graph-based representations. GSC [17] formulates accident anticipation as a graph learning problem with spatio-temporal continuity constraints. Graph(Graph) [18] proposes a nested graph architecture to capture hierarchical agent relationships. DAA-GNN [19] introduces a dynamic attention-augmented graph network that adaptively weights interactions among detected entities. CRASH [20] designs an object-aware module that prioritizes high-risk agents by computing their spatial-temporal relationships.

UString [6] combines relational learning with uncertainty quantification, using Bayesian neural networks to model the stochasticity in agent interactions on its newly collected CCD dataset. AM-Net [21] employs an attention-guided multistream fusion strategy to localize hazardous agents by integrating appearance and motion cues.

### 2.3 Multimodal and Emerging Paradigms

Recent efforts explore richer input modalities and novel learning frameworks. AccNet [22] and CCAF-Net [23] incorporate monocular depth cues to enable 3D-aware scene understanding, fusing RGB and depth features for improved risk prediction. DADA [8] and DADA-2000 [24] frame anticipation as a driver attention prediction task, linking gaze behavior to accident likelihood.

Other innovative directions include reinforcement learning and language grounding. DRIVE [7] models risk prediction as a Markov decision process and uses deep reinforcement learning with visual explanations. CAP [9] establishes a multimodal benchmark for cognitive accident anticipation, integrating behavioral and visual signals. DEDBM [25] fuses dashcam videos with textual accident reports in a dual-branch architecture, enabling cross-modal knowledge transfer. Most recently, WWW [26] leverages large language models (LLMs) to jointly reason about the *what*, *when*, and *where* of potential accidents, marking a shift toward interpretable, language-augmented anticipation systems.

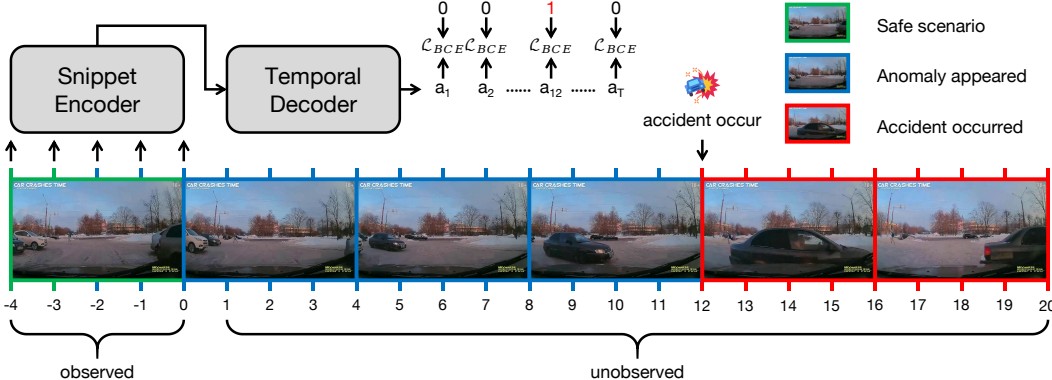

Figure 2: Overview of our encoder-decoder method. We feed the observed frames into the snippet encoder and use the temporal decoder to predict accident scores for unobserved future frames at multiple time steps, where the accident score label is 1 for the accident frame and 0 otherwise.

## 3 Method

### 3.1 Problem Definition

Accident anticipation involves online analysis of dashcam footage to determine whether an accident is likely to occur in the near future. If there is a potential risk of an accident, the system promptly alerts the driver to take precautionary measures, helping to reduce both the likelihood and severity of collisions.

Specifically, given the current frame $f_0$ and past frames $\{f_{-1}, f_{-2}, \dots\}$ captured by the dashcam from the driving scenarios, previous works determine whether an accident will occur in the near future by predicting a risk score $r_0$ at the current time $t_0$. When the risk score $r_0$ exceeds a preset threshold $\tau$, the system will automatically trigger an alert to warn the driver.

In contrast, our method assesses the likelihood of future accidents by predicting a sequence of accident scores $\{a_1, a_2, \dots\}$ of an accident occurring at multiple future time steps $\{t_1, t_2, \dots\}$. When any of the accident scores $a_i (i \geq 1)$ at a certain moment $t_i$ exceeds a preset threshold $\tau$, it indicates that the model predicts an accident will occur at time $t_i$. Consequently, the system will automatically trigger an alert at the current time $t_0$ to warn the driver to take evasive action.

In this task, the model's optimization objective is to maximize the recall rate and earliness of accident anticipation while ensuring that the false alarm rate (FAR) does not exceed an acceptable limit.

### 3.2 Temporal Occurrence Prediction

To anticipate accidents more accurately, we propose a novel paradigm that focuses on predicting a sequence of accident scores of an accident occurring at multiple future time steps, rather than simply outputting a risk score of the current frame, providing a more interpretable and reliable anticipation.

To implement this paradigm, we designed an encoder-decoder model, as shown in Figure 2. The structure details are as follows:

**Snippet encoder.** To better understand the movement of objects in driving scenarios, we take the current frame $f_0$ along with past frames $\{f_{-1}, f_{-2}, \dots, f_{-(S-1)}\}$ as a snippet input to the model, where $S$ is the length of the snippet. Then we employ a 3D CNN instead of architectures of frame-level encoders with RNNs (widely adopted in previous works) as a snippet encoder to simultaneously capture the spatial and temporal information within the snippet. Next, we only apply spatial pooling to the features output by the snippet encoder, in order to preserve their temporal resolution, resulting in features $\{z_0, z_{-1}, \dots, z_{-(S-1)}\}$, where each features correspond to the time steps $\{t_0, t_{-1}, \dots, t_{-(S-1)}\}$, respectively. In this way, the model can not only understand the motion of objects but also establish a one-to-one correspondence between different frames and their corresponding features.

**Temporal decoder.** In order to predict the temporal sequence of accident scores $\{a_1, a_2, \ldots, a_T\}$ of an accident occurring at multiple time steps $\{t_1, t_2, \ldots, t_T\}$, where $T$ is the length of the sequence to predict, in the future based on features $\{z_0, z_{-1}, \ldots, z_{-(S-1)}\}$ extracted by the snippet encoder from frames of time steps $\{t_0, t_{-1}, \ldots, t_{-(S-1)}\}$, we designed a temporal decoder with reference to the transformer decoder [27]. Specifically, to distinguish between different time steps of the temporal sequence in the future, we define $T$ different temporal queries $\{q_1, q_2, \ldots, q_T\}$ to represent $T$ time steps $\{t_1, t_2, \ldots, t_T\}$ in the future following the current timestamp $t_0$ as the reference. Next, we feed the features $\{z_0, z_{-1}, \ldots, z_{-(S-1)}\}$ output from the snippet encoder as the memory into the temporal decoder. Then, we feed temporal queries $\{q_1, q_2, \ldots, q_T\}$ into the temporal decoder to predict the temporal sequence of accident scores $\{a_1, a_2, \ldots, a_T\}$ of an accident occurring at time steps $\{t_1, t_2, \ldots, t_T\}$.

**Sampling strategy.** During training, we randomly sample a continuous segment of $S$ frames from the accident video as the input snippet for the model. Let $f_{-(S-1)}$ be the starting frame of the snippet, then the last frame is $f_0$, which denotes the current frame. To ensure the relevance of training data, snippets are sampled only from frames occurring at or before the accident, while frames after the accident are excluded.

During testing, we adopt a sliding window approach to sample snippets from all available frames in the video, including those before, during, and after the accident. Specifically, we slide a window of $S$ consecutive frames across the entire video sequence with a fixed stride, ensuring comprehensive evaluation of the model's performance over time. This allows the model to make predictions at every time step, reflecting real-world deployment scenarios where the exact timing of an accident is unknown.

**Labeling strategy.** During training, given a snippet across time steps $\{t_0, t_{-1}, \ldots, t_{-(S-1)}\}$ as input, the model outputs the sequence of accident scores $\{a_1, a_2, \ldots, a_T\}$ at multiple time steps $\{t_1, t_2, \ldots, t_T\}$. If the accident occurrence time step $t_A$ falls within this range, i.e., $1 \leq A \leq T$, we assign its label $y_A$ as 1, while setting the labels of all other time steps $y_t$ ($1 \leq t \leq T$, $t \neq A$) to 0, as shown in Figure 2. The model is then trained using the Binary Cross-Entropy (BCE) loss function to optimize its predictions:

$$\mathcal{L}_{BCE} = -\frac{1}{T}\left[ w_+ \log a_A + \sum_{\substack{t=1 \\ t \neq A}}^{T} \log(1 - a_t) \right], \tag{1}$$

where $w_+$ is the weight of the positive one.

## 4    Evaluation Metrics

Previous works [9, 28, 29, 30, 31] primarily used AP (Average Precision), AUC (Area Under the ROC Curve), and TTA (Time-To-Accident) to evaluate accident anticipation methods. However, unlike traditional binary classification tasks, we observe that in real-world applications, excessively high false alarm rates (FAR) can cause unacceptable disturbances to drivers. Therefore, when FAR exceeds a reasonable threshold, comparing recall rates and TTA becomes meaningless. Existing metrics allow FAR to range from 0 to 1, which could lead to suboptimal model selection for practical deployment. To address this, we introduce a threshold $\lambda$ for FAR and only evaluate cumulative recall and TTA when FAR remains below $\lambda$. Furthermore, while current metrics measure overall anticipation capability, they fail to assess performance at specific pre-accident time intervals. Following the approach in [32], we analyze anticipation recall rates at different time intervals before accidents. Finally, we identify limitations in conventional TTA calculation methods and propose an improved alternative. The detailed metrics are described below:

**Area under the ROC curve (AUC).** We employ AUC (Area Under the ROC Curve) to calculate the average recall rate of accident anticipation models under varying false alarm rates (FAR). Notably, when the false alarm rate exceeds a certain level, comparing recall rates across different models becomes meaningless. Therefore, we specifically compute the average recall rate only when the false alarm rate remains below a predefined threshold $\lambda$, denoted as $\text{AUC}^{\lambda}$, as illustrated in Equation 2 and

Figure 4:

$$\text{AUC}^\lambda = \sum_{i=1}^{n} \frac{(\text{TPR}_i + \text{TPR}_{i-1})}{2} \cdot (\text{FPR}_i - \text{FPR}_{i-1}), (\text{FPR}_n \le \lambda), \tag{2}$$

where FPR is equivalent to the false alarm rate (FAR) and TPR is equivalent to the recall rate, $\lambda$ is set to 0.1 by default.

Additionally, to evaluate the capability of the accident anticipation models at different horizons before an accident occurs, we extracted video clips from 0.5s-1.0s, 1.0s-1.5s, and 1.5s-2.0s before the accident as positive samples, while capturing an equal number of 0.5s-long video segments from accident-free driving scenarios as negative samples. We then calculated the $\text{AUC}^\lambda$ for different time intervals, denoted as $\text{AUC}_{0.5s}^\lambda$, $\text{AUC}_{1.0s}^\lambda$, and $\text{AUC}_{1.5s}^\lambda$ (e.g., $\text{AUC}_{1.5s}^\lambda$ represents the model's capability in anticipating accidents 1.5 seconds before they occur). Finally, we computed the model's mean $\text{AUC}^\lambda$ using Equation 3:

$$\text{mAUC}^\lambda = \frac{\text{AUC}_{0.5s}^\lambda + \text{AUC}_{1.0s}^\lambda + \text{AUC}_{1.5s}^\lambda}{3}. \tag{3}$$

**Time-To-Accident (TTA).** We adopt Time-To-Accident (TTA) to evaluate the earliness of the accident anticipation. Specifically, for each frame in an accident video, if the model's predicted anomaly score exceeds a preset threshold $\tau$, an alarm will be triggered, and the time gap (in seconds) between this alarm moment and the actual accident occurrence is recorded as TTA. Generally, as the threshold $\tau$ decreases, both TTA and the false alarm rate (FAR) increase simultaneously. Therefore, we only compute the mean TTA when the false alarm rate remains below $\lambda$, as illustrated in Equation 3:

$$\text{mTTA}^\lambda = \frac{1}{n} \sum_{i=1}^{n} \text{TTA}_i, (\text{FPR}_n \le \lambda). \tag{4}$$

However, as illustrated in Figure 3, the TTA calculation method used in previous works can lead to misleadingly high TTA values. Specifically, when a model raises a false alarm in a safe driving scenario—i.e., it predicts an accident that is not causally linked to any actual future crash—the method still computes TTA based on the time until a subsequent accident (even if unrelated). This results in artificially inflated TTA estimates, sometimes exceeding 3 seconds, despite the prediction being a false alarm.

To address this issue, we propose a revised TTA calculation method: we only compute TTA for alarms triggered after the anomaly appears. This is because the anomaly appearance time is the earliest moment annotated by humans as reliably indicating an impending accident. Alarms issued

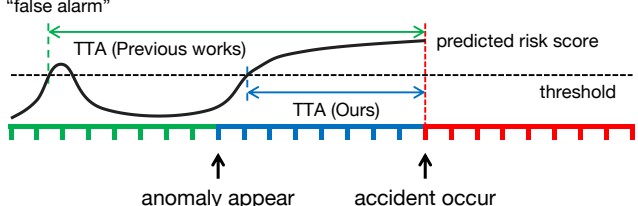

Figure 3: Comparison of the TTA calculation approaches between previous works and our method. We only compute TTA after the anomaly appears, because the moment the anomaly appears is the earliest time annotated by humans as being predictive of the accident. If a model issues an alert earlier than this moment, the perceived risk is typically unrelated to the actual accident that eventually occurs. We consider such alerts as false alarms rather than early warnings. However, previous works did not account for this when calculating TTA, leading to inflated TTA values—sometimes even exceeding 3 seconds.

before this point are considered false alarms, not valid early warnings, as the perceived risk lacks a causal connection to the eventual accident. Under our method, the maximum achievable average TTA is bounded by the average interval between anomaly appearance and accident occurrence in the dataset—1.86 seconds on CAP [9] and 1.66 seconds on DADA [8]. Moreover, if the model fails to predict an accident at all, its TTA is set to 0, ensuring a fair and meaningful evaluation.

# 5 Experiments

## 5.1 Experimental Setup

**Datasets.** We conduct experiments on the MM-AU dataset [33], a large-scale ego-view traffic accident benchmark collected from public sources including existing datasets (CCD [6], A3D [4], DoTA [5], DADA-2000 [24]) and video platforms (YouTube, Bilibili, Tencent), encompassing diverse weather (sunny, rainy, snowy, foggy), lighting (day, night), scenes (highway, tunnel, mountain, urban, rural), and road types (arterial roads, curves, intersections, T-junctions, ramps)—which enables robust evaluation of model generalizability, in contrast to prior works that primarily train on limited datasets like DAD and CCD. MM-AU consists of two subsets: CAP [9] with 11,727 videos (2,195,613 frames) and DADA [8] with 2,000 videos (658,476 frames), both providing annotations for 58 accident categories and temporal labels for key events ("anomaly appear", "accident occur", and "accident end"). We refined and validated the frame rates and annotations for all ego-involved accidents, and selected approximately 20% of the data as the test set; for evaluation, we extract clips from the first frame to the "anomaly appear" frame as negative samples to compute the false alarm rate (FAR), and clips from "anomaly appear" to "accident occur" as positive samples to assess anticipation recall and Time-to-Accident (TTA).

**Implementation details.** We preprocess input videos by resizing each frame to $224 \times 224$ and resampling at 10 FPS, so that each frame corresponds to a 0.1s time interval. During training, we sample snippets only from the period before the accident occurs; during testing, we apply a sliding window over the entire video. Each input snippet consists of $S = 5$ consecutive frames, which are fed into a snippet-level encoder (SlowOnly [34], initialized with ImageNet pre-trained weights) to extract spatiotemporal features. The model then predicts a sequence of accident scores of length $T = 20$, corresponding to future time steps from 0.1s to 2.0s ahead. To decode these scores, we employ a Transformer-based temporal decoder with 2 layers and cosine positional encodings as queries for each future horizon. We optimize the model using SGD with a batch size of 64 on 8 NVIDIA 4090 GPUs. The binary cross-entropy loss is weighted with $w_+ = 10$ for positive samples, and the initial learning rate is set to 0.01, decayed to 10% of its value every 20 epochs over 50 total epochs.

## 5.2 Quantitative Results

**Quantitative comparison.** We evaluate our method against prior approaches and baselines on the CAP [9] and DADA [8] datasets under a unified experimental protocol. For fair comparison, all methods—including CAP [9], DRIVE [7], DSTA [11], and GSC [17]—are trained and tested using the same data splits and evaluation metrics (Section 4). As baselines, we include (1) a ResNet+LSTM architecture that predicts a single per-frame risk score (identical to Experiment I in our ablation study), and (2) a variant of our full model without the temporal decoder.

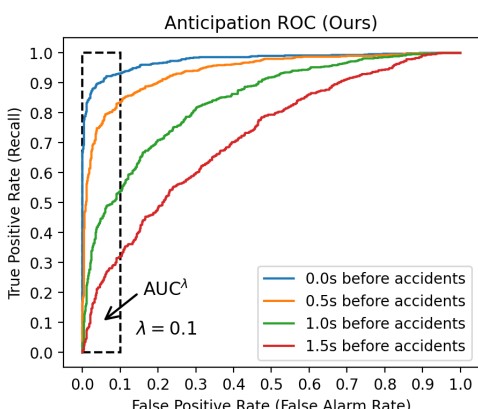

Figure 4: ROC curves of our method on CAP [9] at different accident anticipation horizons: 0.0s, 0.5s, 1.0s, and 1.5s before the accident.

As shown in Table 1, our method achieves significant gains on CAP. At the 0.0s horizon—where the task reduces to precise accident detection—our AUC reaches 0.8381, far surpassing all prior works and the baseline (0.4357). This demonstrates our model's strong capability in recognizing the exact moment of collision. At short-term horizons (0.5s), we obtain an AUC of 0.6747, nearly doubling the best prior result (GSC: 0.4177). The improvement gradually diminishes at longer horizons (1.0s and 1.5s), where risk signals are inherently weaker, yet our method still maintains the highest performance (AUC: 0.3982 and 0.2141, respectively), yielding a mean AUC (mAUC) of 0.4290 and the best mean Time-to-Accident (mTTA = 0.8644 s). These trends are further visualized in the ROC curves of Figure 4, which show consistently higher recall across all false alarm rates, especially near the accident onset.

Table 1: Quantitative results comparison of different methods on the CAP [9] dataset.

| Method | $\text{AUC}^{0.1}$ ↑ | $\text{AUC}^{0.1}_{0.5s}$ ↑ | $\text{AUC}^{0.1}_{1.0s}$ ↑ | $\text{AUC}^{0.1}_{1.5s}$ ↑ | $\text{mAUC}^{0.1}$ ↑ | $\text{mTTA}^{0.1}$ (s) ↑ |
|--------|------|------|------|------|------|------|
| CAP [9] | 0.0421 | 0.0400 | 0.0296 | 0.0373 | 0.0357 | 0.6372 |
| DRIVE [7] | 0.1288 | 0.1167 | 0.1079 | 0.1231 | 0.1159 | 0.3954 |
| DSTA [11] | 0.5593 | 0.3862 | 0.2817 | 0.1913 | 0.2864 | 0.8039 |
| GSC [17] | 0.6093 | 0.4177 | 0.2969 | 0.1994 | 0.3046 | 0.8165 |
| Baseline | 0.4357 | 0.3938 | 0.2770 | 0.1777 | 0.2829 | 0.5389 |
| **Ours** | **0.8381** | **0.6747** | **0.3982** | **0.2141** | **0.4290** | **0.8644** |

Table 2: Quantitative results comparison of different methods on the DADA [8] dataset.

| Method | $\text{AUC}^{0.1}$ ↑ | $\text{AUC}^{0.1}_{0.5s}$ ↑ | $\text{AUC}^{0.1}_{1.0s}$ ↑ | $\text{AUC}^{0.1}_{1.5s}$ ↑ | $\text{mAUC}^{0.1}$ ↑ | $\text{mTTA}^{0.1}$ (s) ↑ |
|--------|------|------|------|------|------|------|
| CAP [9] | 0.0317 | 0.0365 | 0.0670 | 0.0643 | 0.0560 | 0.4964 |
| DRIVE [7] | 0.1005 | 0.0628 | 0.0770 | 0.0885 | 0.0761 | 0.2257 |
| DSTA [11] | 0.4728 | 0.3276 | 0.2207 | 0.1345 | 0.2276 | 0.6952 |
| GSC [17] | 0.5142 | 0.3495 | 0.2382 | 0.1392 | 0.2423 | 0.7034 |
| Baseline | 0.3411 | 0.3046 | 0.2099 | 0.1251 | 0.2132 | 0.4138 |
| **Ours** | **0.7903** | **0.5669** | **0.2877** | **0.1399** | **0.3315** | **0.8848** |

Similar patterns are observed on DADA (Table 2). Our method achieves 0.7903 AUC at 0.0s and 0.5669 at 0.5s, substantially outperforming previous methods. Although the gains at 1.0s–1.5s are more modest, our mAUC (0.3315) and mTTA (0.8848 s) remain the best, confirming the robustness of our approach across datasets. Overall, the results validate that shifting supervision from ambiguous risk labels to precise future accident timing enables more accurate and reliable anticipation, particularly in the critical moments just before a crash.

**Threshold variation.** To evaluate the robustness of our accident anticipation capability under different false alarm constraints, we vary the FAR tolerance threshold $\lambda$—defined as the maximum allowable false alarm rate for computing metrics. When $\lambda = 1$, no constraint is applied, and the evaluation aligns with conventional protocols used in prior works. As $\lambda$ decreases (e.g., to 0.1 or 0.01), metrics are computed only over predictions that satisfy the stricter FAR requirement.

As shown in Table 3, under the practical setting of $\lambda = 0.1$ (i.e., FAR $\leq 10\%$), our method achieves mAUC of 0.4290 on CAP and 0.3315 on DADA, with strong short-term anticipation performance ($\text{AUC}^{0.1}_{0.5s} = 0.6747$ and 0.5669, respectively). Even under a stringent constraint of $\lambda = 0.01$ (FAR $\leq 1\%$), our model retains non-trivial performance, particularly at the 0.5s horizon (AUC = 0.3371 on CAP, 0.1183 on DADA).

Notably, as $\lambda$ decreases, AUC drops more sharply at longer horizons (1.0s–1.5s) than at shorter ones (0.0s–0.5s), indicating that early false alarms are effectively suppressed under tight FAR constraints. This confirms that our model's early predictions are often spurious, while its near-crash anticipation remains reliable—a behavior aligned with real-world safety requirements. The corresponding reduction in mTTA reflects the inherent trade-off between false alarm suppression and anticipation lead time.

## 5.3 Ablation Study

**Temporal occurrence prediction.** Our temporal occurrence prediction (TOP) module replaces the conventional single risk score with a sequence of accident scores over future time steps (0.1s–2.0s), enabling explicit modeling of *when* an accident may occur. As shown in Table 4, adding TOP (Experiment III vs. I) improves performance at the 0.0s horizon (AUC from 0.4357 to 0.5700), but yields limited gains at longer horizons, suggesting that TOP alone—without strong spatiotemporal modeling—struggles to capture early precursors of accidents. However, when combined with the snippet encoder (Experiment IV), TOP contributes significantly to overall anticipation accuracy, confirming that forecasting future accident timing provides more informative supervision than frame-level risk scoring.

Table 3: Quantitative results comparison across different $\lambda$ of our method on the CAP [9] and DADA [8] datasets.

| Dataset | $\lambda$ | $\text{AUC}^\lambda \uparrow$ | $\text{AUC}^\lambda_{0.5s} \uparrow$ | $\text{AUC}^\lambda_{1.0s} \uparrow$ | $\text{AUC}^\lambda_{1.5s} \uparrow$ | $\text{mAUC}^\lambda \uparrow$ | $\text{mTTA}^\lambda$ (s) $\uparrow$ |
|---|---|---|---|---|---|---|---|
| | 1 | 0.9760 | 0.9389 | 0.8377 | 0.7164 | 0.8310 | 1.5908 |
| CAP [9] | 0.1 | 0.8381 | 0.6747 | 0.3982 | 0.2141 | 0.4290 | 0.8644 |
| | 0.01 | 0.7329 | 0.3371 | 0.0882 | 0.0227 | 0.1494 | 0.4394 |
| | 1 | 0.9666 | 0.8946 | 0.7399 | 0.6400 | 0.7582 | 1.4328 |
| DADA [8] | 0.1 | 0.7903 | 0.5669 | 0.2877 | 0.1399 | 0.3315 | 0.8848 |
| | 0.01 | 0.3177 | 0.1183 | 0.0203 | 0.0068 | 0.0484 | 0.5153 |

Table 4: Ablation study on the CAP [9] dataset. TOP: temporal occurrence prediction; SE: snippet encoder.

| Experiment | TOP | SE | $\text{AUC}^{0.1} \uparrow$ | $\text{AUC}^{0.1}_{0.5s} \uparrow$ | $\text{AUC}^{0.1}_{1.0s} \uparrow$ | $\text{AUC}^{0.1}_{1.5s} \uparrow$ | $\text{mAUC}^{0.1} \uparrow$ | $\text{mTTA}^{0.1}$ (s) $\uparrow$ |
|---|---|---|---|---|---|---|---|---|
| I (Baseline) | × | × | 0.4357 | 0.3938 | 0.2770 | 0.1777 | 0.2829 | 0.5389 |
| II | × | ✓ | 0.6027 | 0.5550 | 0.3607 | 0.1931 | 0.3696 | 0.7330 |
| III | ✓ | × | 0.5700 | 0.3432 | 0.2284 | 0.1721 | 0.2479 | 0.4595 |
| **IV (Ours)** | ✓ | ✓ | **0.8381** | **0.6747** | **0.3982** | **0.2141** | **0.4290** | **0.8644** |

**Snippet encoder.** The snippet encoder (SE) processes short clips of consecutive frames to jointly model spatial and temporal dynamics, which is crucial for understanding motion patterns and scene evolution. Comparing Experiment II (SE only) with the baseline (I), SE alone boosts $\text{AUC}_{0.5s}$ from 0.3938 to 0.5550 and mTTA from 0.5389s to 0.7330s. More importantly, when SE is combined with TOP (Experiment IV), the model achieves the best results across all metrics: $\text{AUC}^{0.1} = 0.8381$, mAUC $= 0.4290$, and mTTA $= 0.8644$s. This demonstrates that SE and TOP are highly complementary—SE provides rich spatiotemporal context, while TOP leverages precise temporal supervision to produce well-calibrated anticipation outputs.

# 6 Conclusion

In this work, we propose a novel accident anticipation paradigm that shifts the prediction target from ambiguous per-frame risk scores to directly estimating accident scores at multiple future time steps (e.g., 0.1s–2.0s ahead), leveraging precisely annotated accident occurrences as supervision. Our method employs a snippet encoder and a Transformer-based temporal decoder to jointly model spatiotemporal dynamics and enable online anticipation. Furthermore, we introduce a practical evaluation protocol that reports recall and Time-to-Accident (TTA) only under acceptable false alarm rates, aligning metrics with real-world deployment needs. Experiments show that our approach achieves state-of-the-art performance, particularly in the critical moments just before a crash.

**Limitations.** While our method significantly improves anticipation accuracy near the accident onset, its performance at longer horizons (e.g., >1.0s) remains limited, indicating challenges in capturing subtle early precursors. Additionally, even under constrained false alarm rates, spurious alerts can still occur in complex scenes, which may affect user trust. These issues point to key directions for future work.

**Potential societal impacts.** Our system has the potential to enhance road safety by providing timely warnings. However, over-reliance on automated alerts might reduce driver vigilance. Careful human-in-the-loop design and user education are essential to maximize safety benefits while mitigating behavioral risks.

## Acknowledgments and Disclosure of Funding

This work was supported by the National Natural Science Foundation of China (Grant No. 62471344), the Zhongguancun Academy (Project No. 20240304), and the CCF-DiDi GAIA Collaborative Research Funds for Young Scholars.

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

# A Qualitative Results

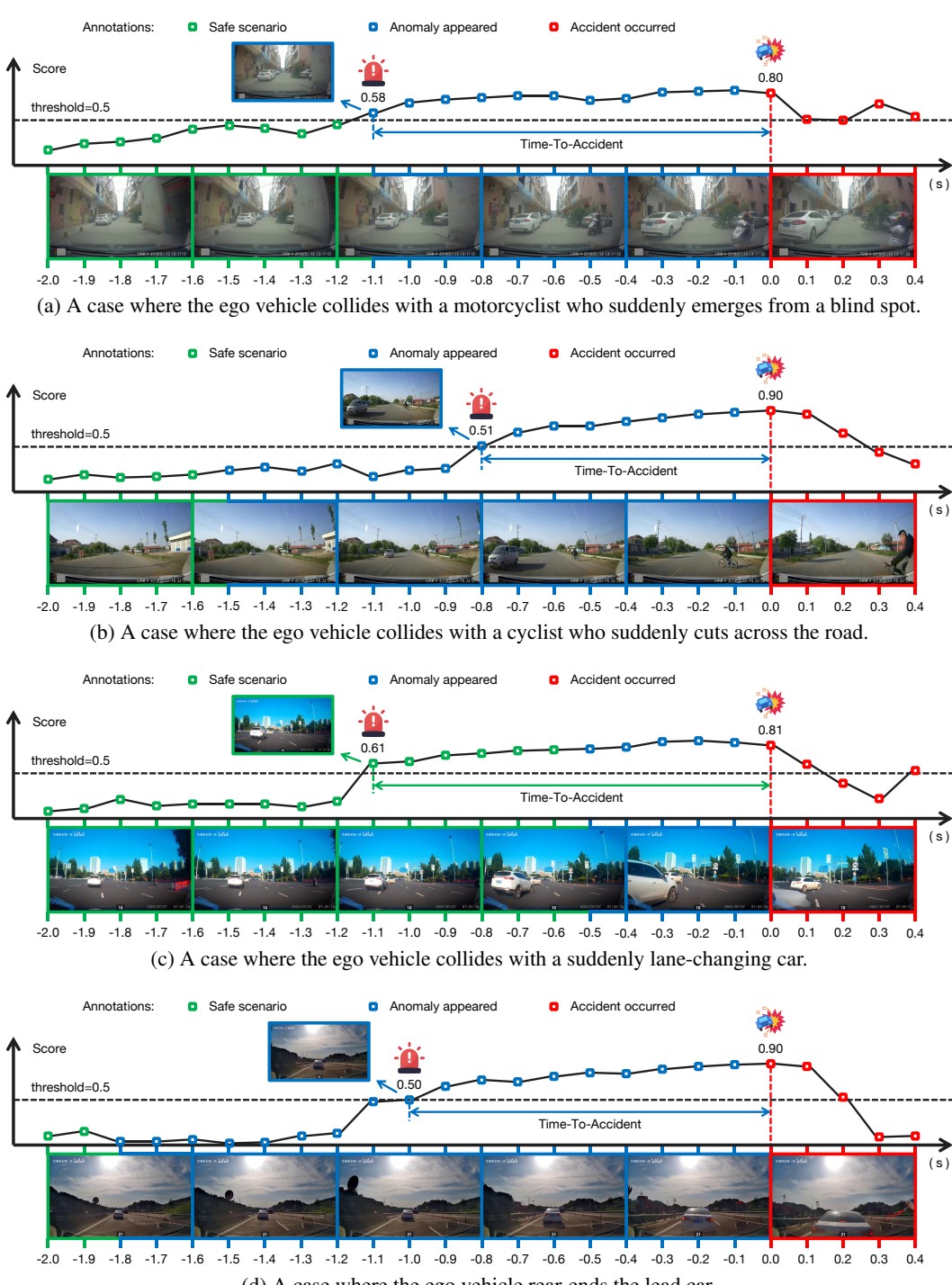

(a) A case where the ego vehicle collides with a motorcyclist who suddenly emerges from a blind spot.

(b) A case where the ego vehicle collides with a cyclist who suddenly cuts across the road.

(c) A case where the ego vehicle collides with a suddenly lane-changing car.

(d) A case where the ego vehicle rear-ends the lead car.

Figure 5: Qualitative results on CAP [9]. Each case shows the trend of the maximum accident score predicted over future time steps; an alarm is triggered if this maximum exceeds the threshold.

We present the qualitative results of our method on the CAP dataset [9] in Figure 5, where different colors denote the temporal annotations in the dataset. Four distinct cases are demonstrated: (a) and (b) involve ego-vehicle collisions with vulnerable road users, while (c) and (d) involve collisions with other vehicles.

We trigger an alarm if any accident score within the 2.0s prediction horizon exceeds a predefined threshold (e.g., 0.5). In cases (a) and (b), the model issues an alert immediately when a motorcyclist emerges from a blind spot or a cyclist begins to cut across the road. In cases (c) and (d), the model accurately responds to sudden lane changes and abrupt braking of the lead vehicle, demonstrating reliable anticipation under diverse hazardous scenarios.

The average Time-to-Accident (TTA) across these four cases is 1.0s, consistent with the typical duration between anomaly onset and collision in real-world accidents. Notably, previously reported TTAs exceeding 3 seconds in prior works [7, 9] stem from flawed calculation methods that count false alarms far before the anomaly as valid early predictions—rather than genuine long-term anticipation capability. This underscores the necessity of our revised TTA metric.

Furthermore, we observed inconsistencies in the dataset's "anomaly appear" annotations. For instance, cases (b) and (d) were labeled too early, while case (c) was labeled too late. Such subjectivity introduces noise when using anomaly onset as a training or evaluation boundary.

