# OpenReview forum: "Accident Anticipation via Temporal Occurrence Prediction"
_NeurIPS.cc/2025/Conference — NeurIPS 2025 poster_

### Official Review · Reviewer_o9V3 · 2025-06-18

**Clarity:** 2
**Significance:** 2
**Originality:** 2
**Rating:** 5
**Confidence:** 3

**Summary:**

This paper introduces a method for traffic accident anticipation in terms of spatiotemporal snippet encoder and multiple time-stamp anticipation. Additionally, for the purpose of practical traffic application usage, this paper also considers to assign improved evaluation protocols with recall rate and TTA. This paper shows the effectiveness of performance and anticipation earliness in the experimental section.

**Questions:**

- In Tables 1 & 2, the scores show improved performance, but it is unclear whether this improvement is due to the proposed method itself or other factors. If possible, could you clarify, by means of a correspondence table, what detection methods, architectures, or other factors were used for the proposed vs. the conventional methods [7, 8, 9]? The reviewer also believes that these details should be clearly documented in the paper. If the reviewer has overlooked this, he sincerely apologizies. Please point out where the paper described the points?

**Ethical Concerns:**

["NO or VERY MINOR ethics concerns only"]

**Final Justification:**

The reviewer thanks the authors for a helpful information and great efforts. After the rebuttal phase, the reviewer has a better understanding on the proposed method. This has a contribution to the topic of traffic accident anticipation. Hopefully the proposed method can be widely shared in the research community and employed diverse traffic applications.

**Limitations:**

Yes

**Paper Formatting Concerns:**

Nothing about the paper formatting concerns.

**Quality:**

2

**Strengths And Weaknesses:**

- S1. The proposed method is well-deserved and well-motivated in the topic of traffic accident anticipation. At the beginning, this approach follows the conventional manners such as AUC, TTA, and FPS. On top of that, the proposed method additionally proposes snippet encoder and multiple time-span (from 0.1 to 2.0 seconds) anticipation.

- S2. The proposed method has been compared to the conventional approaches including recent works including the reference [7,8,9]. Totally, the propsed method serves better results in terms of performance and earliness in the real-time traffic accident anticipation.

- S3. Basically the paper is written properly. For example, the figures are put on the right locations and included good presentation for describing the methodology and results.

- W1. The paper describes as "Existing methods typically predict frame-level anomaly scores as risk indicators," in l.2-3, and "For example, a branch of works [3, 4, 5, 6, 7] regards the risks as 1 for frames within a fixed interval (e.g. 2.0s) before the annotated accident timestamp and 0 for frames sampled from safe driving scenarios," in l.26-28. However, for example, it seems the paper [Kumamoto+, WACV25] assigns Transformer model and anticipate traffic accidents from multi-frame input. On the other hand, the paper [Suzuki+, CVPR18] and following papers have been considered the traffic accident anticipation with flexible frame lengths in the adaptive loss function. Therefore, the descriptions in l.2-3 & l.26-28 are slightly overclaiming according to what they do in the conventional works.

[Kumamoto+, WACV25] Y. Kumamoto et al. "AAT-DA: Accident Anticipation Transformer with Driver Attention," WACV 2025.

[Suzuki+, CVPR18] T. Suzuki et al. "Anticipating Traffic Accidents with Adaptive Loss and Large-scale Incident DB," CVPR 2018.


- W2. Lacking some important citations. Other than the above mentioned [Kumamoto+, WACV25] and [Suzuki+, CVPR18], this paper has no citation of e.g., [Karim+, TITS22]. This relates to the paper quality.

[Karim+, TITS22] M. Karim et al. "A Dynamic Spatial-Temporal Attention Network for Early Anticipation of Traffic Accidents," IEEE Transactions on Intelligent Transportation Systems, 2022.

- W3. Certainly, some proposed approaches are somewhat new such as the snippet encoder and multiple timestamps. However, these are not conceptually very novel in the topic of traffic accident anticipation. Although the performance has been improved by comparing to the conventional approaches, I am concerning its coming from the novel detection methods and architectural changes.

---

> ### Author Rebuttal · Authors · 2025-07-26
>
> Dear Reviewer o9V3,
>
> Thank you for your detailed review and valuable feedback on our submission "Accident Anticipation via Temporal Occurrence Prediction". We appreciate your recognition of our method's strong motivation, its effective comparison with conventional approaches, and the overall clarity of our paper's presentation.
>
> We have carefully considered your comments, especially regarding the originality claims, missing citations, and the need for clearer ablation studies, and we're committed to addressing them to improve our manuscript.
>
> ## **Response to Weaknesses and Questions:**
> **W1. Clarity on Our Claims:**
> We sincerely apologize for any ambiguity or confusion caused by our claims. We have provided clarifications for each of our claims and guarantee that these will be revised in the accepted version of the paper to prevent any future misunderstanding for readers.
> - **"Existing methods typically predict frame-level anomaly scores as risk indicators," in l.2-3:**
> We would like to clarify that our contribution **does not lie in the use of multi-frame input, but in predicting multiple timestamp risk sequence**. We fully acknowledge that many existing methods, including [Kumamoto+, WACV’25], utilize multi-frame inputs and powerful temporal encoders such as Transformers to extract temporal features. However, our claim means that prior works predict **a single anomaly score** for each frame, while our method predicts **a future sequence of anomaly scores** for each frame, such as the likelihood of an accident occurring at +0.1s, +0.2s, ..., up to +2.0s. This allows our model to explicitly capture the temporal evolution of accident risk over a future horizon.
>
> - **"For example, a branch of works [3, 4, 5, 6, 7] regards the risks as 1 for frames within a fixed interval (e.g. 2.0s) before the annotated accident timestamp and 0 for frames sampled from safe driving scenarios," in l.26-28:**
> We would like to clarify that while the paper [Suzuki+, CVPR18] and following papers do introduce adaptive loss functions—typically by applying penalty weights that decay with temporal distance from the accident frame—they still assign a binary label of 1 to all frames within a fixed interval (e.g., 2.0s before the accident). In other words, although the adaptive loss functions reduce the penalty for frames farther from the accident, the model is still optimized toward **predicting a value close to 1 for all positive frames**, which may not reflect the actual uncertainty or gradual buildup of risk over time. In contrast, we argue that only the accident timestamp itself can be precisely labeled, while risk at earlier frames is uncertain and should be predicted in a temporally resolved and probabilistic manner, rather than assigned a flat label. Our method addresses this by **predicting a distribution of future accident probabilities**, which better captures the evolving nature of risk.
>
> **W2. Lacking Important Citations:**
> Thank you for bringing these important missing citations to our attention. We agree that including relevant prior work is crucial for paper quality and proper contextualization. We will revise our manuscript to include comprehensive discussions of:
>     * **[Kumamoto+, WACV25]**: We will discuss their use of Transformer models and driver attention for accident anticipation,
>     * **[Suzuki+, CVPR18]**: We will include their work on adaptive loss functions and large-scale incident databases.
>     * **[Karim+, TITS22]**: We will discuss their Dynamic Spatial-Temporal Attention Network, acknowledging its contribution to early anticipation and its approach to spatio-temporal modeling.
>     * **Other works**: We will investigate the latest work and incorporate it into our related work for discussion.
>
> **W3. Novelty and the Causes of Performance Improvement:**
> - **Novelty:**  The key novelty of our work lies in the novel paradigm of predicting the temporal occurrence of an accident. Specifically, we move from predicting a single risk score for each frame to predicting a temporal sequence of future accident probabilities for each frame. This formulation allows the model to better capture the evolving nature of risk in dynamic driving scenes.
> - **Causes of Performance Improvement:**
> Our method does not contain object detection modules, nor does it depend on complex architectural modifications. Since two of our contributions are the novel paradigm of predicting the temporal occurrence of an accident and the snippet encoder, we have highlighted their effectiveness in our ablation studies (see Sec. 5.3).
>
> As shown in Table 4, our main contribution, "predicting the temporal occurrence" (TOP), brings performance improvement to mAUC$^{0.1}$ (0.0201) and mTTA$^{0.1}$ (0.0510), especially to AUC$_{0.0s}^{0.1}$ (0.1715).
>
> Our second contribution, "snippet encoder" (SE), also brings performance improvement to mAUC$^{0.1}$ (0.1378) and mTTA$^{0.1}$ (0.3006), especially to AUC$_{0.0s}^{0.1}$ (0.2090), while experiments I, II, V, VI utilized RNNs to aggregate temporal features.
>
> Based on the above, these ablation experiments demonstrate that our proposed novel contributions are effective and can lead to performance improvements.
>
> Table 4: Ablation study on the CAP-DATA [8] dataset, where TOP denotes the temporal occurrence prediction, SE denotes the snippet encoder, and AAA denotes the anomaly appearance annotations.
>
> | Experiment     | TOP | SE | AAA | AUC$_{0.0s}^{0.1} \uparrow$ | AUC$_{0.5s}^{0.1} \uparrow$ | AUC$_{1.0s}^{0.1} \uparrow$ | AUC$_{1.5s}^{0.1} \uparrow$ | mAUC$^{0.1} \uparrow$ | mTTA$^{0.1}$ (s) $\uparrow$ |
> |-|-|-|-|-|-|-|-|-|-|
> | I (Baseline)   | ✕   | ✕  | ✕   | 0.4357                    | 0.3951                    | 0.2770                    | 0.1777                    | 0.2829                | 0.5389                      |
> | II             | ✕   | ✕  | ✓   | 0.5151                    | 0.3951                    | 0.2353                    | 0.1491                    | 0.2598                | 0.5551                      |
> | III            | ✕   | ✓  | ✕   | 0.6027                    | 0.5550                    | 0.3607                    | 0.1931                    | 0.3696                | 0.7330                      |
> | IV             | ✕   | ✓  | ✓   | 0.7251                    | 0.6280                    | 0.3671                    | 0.1949                    | 0.3967                | 0.8046                      |
> | V              | ✓   | ✕  | ✕   | 0.5700                    | 0.3432                    | 0.2284                    | 0.1721                    | 0.2479                | 0.4595                      |
> | VI             | ✓   | ✕  | ✓   | 0.6827                    | 0.4395                    | 0.2510                    | 0.1584                    | 0.2830                | 0.5790                      |
> | VII            | ✓   | ✓  | ✕   | 0.8381                    | 0.6747                    | **0.3982** | **0.2141** | 0.4290                | 0.8644                      |
> | **VIII (Ours)**    | ✓   | ✓  | ✓   | **0.8739** | **0.6979** | 0.3892                    | 0.2014                    | **0.4295** | **0.9327** |
>
> **Questions: Clarifying Performance Improvement Causes:**
> Since we have clarified that our contributions could cause performance improvement in the above "W3" with ablation studies in Table 4, we will clarify the difference in implementation details between the conventional methods and ours in this response.
>
> We list all the components used in previous works (including text input, driver attention, object detector, spatial encoder, GCNs, RNNs) in the following Table. Compared with previous works using a spatial encoder and RNNs to understand spatial and temporal features, we capture spatial and temporal features simultaneously by a snippet encoder, SlowOnly (a very vanilla backbone in video recognition). Additionally, we did not use any other structures (like object detectors and GCNs) or auxiliary information (like text input and driver attention) to ensure simplicity of implementation. Despite this simplicity, we also achieve SOTA performance (see Rebuttal 2. for Reviewer frVz) within fair comparison and Kaggle competition. This demonstrates that the improvement is due to the proposed method itself rather than novel detection methods and architectural changes.
>
> |Method|Text Input|Driver Attention|Object Detector|Spatial Encoder|GCNs|RNNs|TOP|mAUC$^{0.1}$ ↑|mTTA$^{0.1}$ (s) ↑|
> |:-|:-|:-|:-|:-|:-|:-|:-|:-:|:-:|
> |DRIVE [Bao+, ICCV21]|✕|✓|✕|VGG-16-based MLNet|✕|✕|✕|0.1159|0.3954|
> |CAP [Fang+, ITSM24]|✓|✓|✕|CNN|✓|GRU|✕|0.0357|0.6372|
> |DSTA [Karim+, TITS22]|✕|✕|Cascade R-CNN|ResNet-101|DSN|GRU|✕|0.2864|0.8039|
> |GSC [Wang+, TIV23]|✕|✕|Faster R-CNN|ResNet-101|✓|LSTM|✕|0.3046|0.8165|
> |Baseline|✕|✕|✕|Resnet-50|✕|LSTM|✕|0.2829|0.5389|
> |Ours|✕|✕|✕|Snippet Encoder (SlowOnly)|✕|✕|✓|0.4295|0.9327|
>
> [Bao+, ICCV21] W. Bao et al., “Drive: Deep reinforced accident anticipation with visual explanation”, in ICCV, 2021.
>
> [Fang+, ITSM24] J. Fang et al., “Cognitive accident prediction in driving scenes: A multimodality benchmark”, in ITSM, 2024.
>
> [Karim+, TITS22] M. M. Karim et al., “A dynamic spatial-temporal attention network for early anticipation of traffic accidents”, in TITS, 2022.
>
> [Wang+, TIV23] T. Wang et al., “GSC: A graph and spatio-temporal continuity based framework for accident anticipation”, in TIV, 2023.

---

> > ### Comment · Reviewer_o9V3 · 2025-08-07
> > **Official Comment by Reviewer o9V3**
> >
> > W1.
> > > We would like to clarify that our contribution does not lie in the use of multi-frame input, but in predicting multiple timestamp risk sequence.
> >
> > Sorry for my misunderstanding. Its the multiple timestamp on behalf of the multi-frame input. The following response regarding W1 is also addressing the reviewer's concerns.
> >
> > W2.
> > > We agree that including relevant prior work is crucial for paper quality and proper contextualization. We will revise our manuscript to include comprehensive discussions...
> >
> > Thank you so much for the consideration. It will be helpful for broder readers not only the reviewer.
> >
> >
> > W3.
> > Thank you so much for the detailed response. Especially, the updated table "Questions: Clarifying Performance Improvement Causes" is really understandable for the reviewer. According to the table, does this mean that the proposed method (Snippet Encoder (SlowOnly)) has the greatest effect? Also, how should we interpret the impact when it is combined with other factors not limited to a single factor?
> >
> >
> > Throughout the author response, the reviewer is leaning to the positive side in the paper rating.

---

> > > ### Author Response · Authors · 2025-08-08
> > >
> > > Dear Reviewer o9V3,
> > >
> > > Thank you for your recognition of our work.
> > >
> > > **Q1:** According to the table, does this mean that the proposed method (Snippet Encoder (SlowOnly)) has the greatest effect?
> > >
> > > **R1:** While the Snippet Encoder (SE) indeed provides great improvement by jointly capturing spatial and temporal features of driving scenarios, the experiments do not indicate that SE alone contributes the most to the overall performance improvement. Specifically, with only SE (Exp. 6), our model obtains an mTTA of 0.7330, which is still below previous methods such as DSTA [Karim+, TITS22] and GSC [Wang+, TIV23], with mTTA values of 0.8039 and 0.8165, respectively. Crucially, by building upon the robust spatiotemporal features extracted by SE, our proposed Temporal Occurrence Prediction (TOP) training paradigm effectively captures the evolving likelihood of accidents over time. This leads to a significant further improvement of mTTA to 0.8644 (Exp. 7), clearly surpassing all prior state-of-the-art methods.
> > >
> > > **Q2:** How should we interpret the impact when it is combined with other factors, not limited to a single factor?
> > >
> > > **R2:** Each component (TOP, SE, AAA) brings a unique effect, and their combination can yield synergistic improvements. In our method, SE lays a strong spatial-temporal feature foundation, which enables other modules, such as TOP, to operate effectively. With such high-quality features, TOP can accurately predict multi-timestamp risk sequences. Ultimately, when these modules are jointly applied, their complementary strength lead to a substantial performance gain (mTTA = 0.9327) (Exp. 8), demonstrating that the combination is more powerful than any single component in isolation.
> > >
> > >
> > > | Exp. | Method | Text Input | Driver Attention | Object Detector | Spatial Encoder | GCNs | RNNs | mAUC$^{0.1}$ ↑ | mTTA$^{0.1}$ (s) ↑ |
> > > |:-|:-|:-:|:-:|:-:|:-:|:-:|:-:|:-:|:-:|
> > > |1| DRIVE [Bao+, ICCV21] | ✕ | ✓ | ✕ | VGG-16-based MLNet | ✕ | ✕ | 0.1159 | 0.3954 |
> > > |2| CAP [Fang+, ITSM24] | ✓ | ✓ | ✕ | CNN | ✓ | GRU | 0.0357 | 0.6372 |
> > > |3| DSTA [Karim+, TITS22] | ✕ | ✕ | Cascade R-CNN | ResNet-101 | DSN | GRU | 0.2864 | 0.8039 |
> > > |4| GSC [Wang+, TIV23] | ✕ | ✕ | Faster R-CNN | ResNet-101 | ✓ | LSTM | 0.3046 | 0.8165 |
> > > |5| Baseline | ✕ | ✕ | ✕ | Resnet-50 | ✕ | LSTM | 0.2829 | 0.5389 |
> > > |6| Ours (SE) | ✕ | ✕ | ✕ | Snippet Encoder (SlowOnly) | ✕ | ✕ | 0.3696 | 0.7330 |
> > > |7| Ours (SE + TOP) | ✕ | ✕ | ✕ | Snippet Encoder (SlowOnly) | ✕ | ✕ | 0.4290 | 0.8644 |
> > > |8| **Ours** (SE + TOP + AAA)| ✕ | ✕ | ✕ | Snippet Encoder (SlowOnly) | ✕ | ✕ | **0.4295** | **0.9327** |
> > >
> > > [Bao+, ICCV21] W. Bao et al., “Drive: Deep reinforced accident anticipation with visual explanation”, in ICCV, 2021.
> > >
> > > [Fang+, ITSM24] J. Fang et al., “Cognitive accident prediction in driving scenes: A multimodality benchmark”, in ITSM, 2024.
> > >
> > > [Karim+, TITS22] M. M. Karim et al., “A dynamic spatial-temporal attention network for early anticipation of traffic accidents”, in TITS, 2022.
> > >
> > > [Wang+, TIV23] T. Wang et al., “GSC: A graph and spatio-temporal continuity based framework for accident anticipation”, in TIV, 2023.

---

> ### Author Response · Authors · 2025-08-05
>
> Dear Reviewer o9V3,
>
> We hope that we have addressed your questions in our rebuttal. Do you have any further questions? We would be delighted to have a discussion.

---

### Official Review · Reviewer_nKX6 · 2025-06-23

**Clarity:** 2
**Significance:** 3
**Originality:** 3
**Rating:** 5
**Confidence:** 3

**Summary:**

The paper proposes a novel accident anticipation paradigm that shifts the prediction target from frame-level anomaly scores to the probability of future accident occurrences at multiple time steps, enabling more precise supervision and more interpretable predictions. Specifically, it employs a snippet encoder and a Transformer-based temporal decoder, and introduces a new evaluation protocol that considers false alarm rates.

**Questions:**

1.	One of the main contributions of the paper is the shift “from frame-level anomaly scores to the probability of future accident occurrences at multiple timestamps.” **However, is there a clear distinction between ‘scores’ and ‘probability’?** The authors appear to use “score” and “probability” interchangeably throughout the paper (e.g., lines 136–138 on page 4: “we propose…**predicting the probabilities** …rather than simply outputting the **anomaly scores** of…”; lines 287–288 on page 7: “For our baseline, we adopt a CNN+RNN architecture to predict an **anomaly score** and …”), yet on lines 332–334 of page 8 they explicitly state: “This validates that explicit **probability prediction** at future time steps provides more reliable signals than **anomaly score-based** methods.”

2.	The authors repeatedly state that “when FAR exceeds a reasonable threshold, comparing recall rates and TTA becomes meaningless.” **However, they should report the actual false alarm rates (FAR) of existing methods and compare them with their own approach**, especially since this is a key motivation of the paper. Simply stating this claim without quantitative evidence weakens the argument.

3.	In paper [1], it is pointed out that “Intuitively, the penalty of failing to anticipate an accident at a frame very close to the accident should be higher than the penalty at a frame far away from the accident.” That is, as a probability prediction task, the predicted probability should increase as it approaches the accident frame. However, in this paper, the same loss function (Equation 1) is applied to both positive and negative samples. Given the temporal nature of the task, is there any mechanism in the loss function to ensure that frames closer to the accident are predicted with higher probabilities? Additional explanation on this point would strengthen the paper.

4. Do the authors plan to release the code and pretrained models upon acceptance to support reproducibility of the results?

[1] Anticipating Accidents in Dashcam Videos.

Note：If the authors can address the above issues, I would consider raising my score.

**Ethical Concerns:**

["NO or VERY MINOR ethics concerns only"]

**Final Justification:**

The author’s explanation is convincing and has resolved all my questions. The author has also promised to provide open-source code to ensure the reproducibility of the results.

**Limitations:**

yes

**Quality:**

3

**Strengths And Weaknesses:**

**Strengths**:
This paper proposes a novel accident anticipation paradigm and introduces a new evaluation protocol that accounts for false alarm rates, thereby improving the accuracy and timeliness of accident prediction.

**Weaknesses**:
The writing in the paper lacks clarity in some parts; please refer to the “Questions” section for details

---

> ### Author Rebuttal · Authors · 2025-07-26
>
> Dear Reviewer nKX6,
>
> Thank you for your valuable time and insightful feedback on our submission "Accident Anticipation via Temporal Occurrence Prediction". We appreciate your recognition of our novel accident anticipation paradigm and the introduction of a new evaluation protocol that accounts for false alarm rates. Your comments are crucial for improving the clarity and rigor of our work.
>
> We have carefully considered each of your points and would like to address them as follows:
>
> ## **Response to Questions:**
> **1. Distinction Between "Score" and "Probability":**
> Thank you for pointing out this important issue. We would like to clarify that in our paper, “anomaly score” and “probability” refer to the same concept—a quantitative value indicating the likelihood of an accident. We will revise the manuscript to standardize the terminology for clarity.
>
> - **Clarification of the main contribution:**
> The key distinction we aim to highlight is not the difference between “score” and “probability,” but rather the shift from predicting **a single score/probability** to predicting **a temporal sequence of such scores/ probabilities**. Specifically, previous methods predict one anomaly score/risk probability per frame, indicating the general likelihood of an accident. In contrast, for each frame, our method predicts a sequence of future probabilities, such as the likelihood of an accident occurring at +0.1s, +0.2s, ..., up to +2.0s. This allows our model to explicitly capture the temporal evolution of accident risk over a future horizon.
>
> We appreciate the reviewer’s observation and will ensure that this distinction is clearly articulated in the revised manuscript.
>
> **2. Reporting False Alarm Rates (FAR) of Existing Methods:**
> Thank you for the valuable comment. We agree that quantitative evidence is essential to support our claim regarding the limitations of existing methods under high false alarm rates (FAR). In practice, the false alarm rate is threshold-dependent, making direct comparisons between methods at a fixed threshold inappropriate. Consequently, to ensure a fair comparison, we follow a consistent evaluation protocol: we report the FAR of each method when it achieves 80% early warning recall rate. The results of this evaluation are summarized in Table 1.
>
> Table 1: False Alarm Rate (FAR) of each method when achieving 80% early warning recall.
> | Method | Drive [1] | CAP [2] | DSTA [3] | GSC [4] | Baseline | Ours |
> |:-|:-:|:-:|:-:|:-:|:-:|:-:|
> | FAR $\downarrow$ | 0.59 | 0.47 | 0.23 | 0.19 | 0.25 | **0.01** |
>
> [1] W. Bao et al., “Drive: Deep reinforced accident anticipation with visual explanation”, in ICCV, 2021.
>
> [2] J. Fang et al., “Cognitive accident prediction in driving scenes: A multimodality benchmark”, in ITSM, 2024.
>
> [3] M. M. Karim et al., “A dynamic spatial-temporal attention network for early anticipation of traffic accidents”, in TITS, 2022.
>
> [4] T. Wang et al., “GSC: A graph and spatio-temporal continuity based framework for accident anticipation”, in TIV, 2023.
>
> Our method ("Ours") achieves a remarkably low False Alarm Rate of 0.01. This is significantly lower than all other compared methods: Drive [1] (0.59), CAP [2] (0.47), DSTA [3] (0.23), GSC [4] (0.19), and the Baseline (0.25). This result strongly demonstrates our method's superior capability in controlling false alarms. In real-world applications, a low False Alarm Rate is crucial because it indicates that the system can more accurately identify genuine accident risks without frequently issuing erroneous alerts. An excessively high false alarm rate (like 0.59 and 0.47) can lead to user fatigue and distrust, thereby diminishing the system's practical utility. That's why we proposed a new evaluation protocol that explicitly measures recall rate and Time-to-Accident (TTA) *only under acceptable false alarm rates*. We will include this evaluation and the corresponding results in the revised manuscript to substantiate our motivation and improve clarity.
>
> **3. Temporal Exponential Decay for Penalty Weights of Loss Function:**
> Thank you for your thoughtful question regarding the temporal weights of our loss function, especially considering the idea that earlier anticipations might warrant lower penalties. Such mechanisms (e.g., [1]'s implementation) give a lower penalty weight to the loss for earlier anticipations of accident videos in a temporal exponential decay manner ($e^{-\max(0,y-t)}$), while the labels remain 1 for all frames in the accident video, as shown in Equation (12).
>
> $$L_p(\{a_t\}) = \sum_t -e^{-\max(0,y-t)} \log(a_t^0), \quad (12)$$
>
> Similarly, we can apply this mechanism (Exponential Weights) to our method. We conducted ablation studies of this mechanism on the Baseline method (in the way of predicting an anomaly score) and our method (in the way of predicting a sequence of probabilities at multiple timestamps in the future), as shown in the following table. This mechanism could bring a large improvement to the Baseline method of late anticipations (e.g., AUC$^{0.1}_{0.5s}$). However, it only brings a subtle improvement to our method. This is because we model the supervision signal of different timestamps before the accident independently (see Rebuttal 2 for Reviewer TYgU), while previous works model them as the same "anomaly score" (the uncorrected label of early anticipations may degrade the ability of late anticipations). Therefore, this experimental result proves that our method is more robust to the penalty weights of the loss function.
>
> Table 2. Impact of Exponential Temporal Weights on the baseline and our method.
> | Method | Exponential Weights | AUC$^{0.1}_{0.0s}$ ↑ | AUC$^{0.1}_{0.5s}$ ↑ | AUC$^{0.1}_{1.0s}$ ↑ | AUC$^{0.1}_{1.5s}$ ↑ | mAUC$^{0.1}$ ↑ | mTTA$^{0.1}$ (s) ↑|
> | :- | :- | :-: | :-: | :-: | :-: | :-: | :-: |
> | Baseline |✕| 0.3897 | 0.3042 | 0.2359 | 0.1842 | 0.2414 | 0.5452 |
> | Baseline |✓| 0.5142 | 0.3967 | 0.2463 | 0.1652 | 0.2694 | 0.5427 |
> | Ours  |✕| 0.8381 | 0.6747 | 0.3982 | 0.2141 | 0.4290 | 0.8644 |
> | Ours  |✓| 0.8538 | 0.6853 | 0.3931 | 0.2057 | 0.4280 | 0.9036 |
>
> **4. Code and Pretrained Model Release:**
> Yes, we **plan to release our code, pretrained models, and detailed documentation upon acceptance** to support full reproducibility. Our method achieves state-of-the-art (SOTA) performance on both the MM-AU and Nexar datasets. More importantly, we propose a more reasonable and standardized evaluation protocol for accident anticipation. By releasing our implementation along with strong SOTA baselines, we aim to facilitate fair comparison and accelerate progress in future research on this task.

---

> > ### Comment · Reviewer_nKX6 · 2025-08-03
> > **Rebuttal Reply**
> >
> > The authors have addressed my previous confusion, but regarding the TTA calculation. I have a new question.
> >
> > During the inference process, the authors obtained a temporal probability sequence {$ {p_{C+1}, . . . , p_{C+U}} $} of an accident occurring at timestamps {$ {t_{C+1}, . . . , t_{C+U}} $}. According to my understanding, the safe driving scenario, anomaly appearance, and accident occurrence all fall within {$ {t_{C+1}, . . . , t_{C+U}} $}. As long as the probability of any frame exceeds the threshold and triggers an alarm, it would alert the driver. However, the authors state that they "only consider alarms triggered after the anomaly occurs and compute the TTA accordingly." In practical inference applications without dataset annotations, how can we determine that the alarm occurred after the "anomaly appear" moment rather than only informing the driver at the "second alarm"?

---

> > > ### Author Response · Authors · 2025-08-04
> > >
> > > Dear Reviewer nKX6,
> > >
> > > Thank you for your insightful question. **In practical inference applications**, our system doesn't rely on "anomaly appearance" annotations. It issues an immediate warning the moment the predicted probability for any frame exceeds the threshold, just as you described.
> > >
> > > However, for a fair and meaningful **evaluation** on existing dataset benchmarks, we adopt the following procedure to compute the metrics: (1) if the alarm triggers earlier than the "anomaly appearance", we regard this as a false alarm. (2) if the alarm triggers later than the "anomaly appearance", this alarm is more likely to be relevant to the accident; thus, we calculate the TTA between the first alarm after the "anomaly appearance" and the accident. This ensures that models with both a low false alarm rate and a high TTA are favored, aligning better with real-world requirements.
> > >
> > > In contrast, prior methods such as CAP [2] use the original TTA definition, which reports a TTA of 4.648s on 5-second video clips, where the average time from anomaly appearance to accident is 1.86s. Such methods tend to raise alarms when no anomaly appears, making them not applicable for real-world applications.
> > >
> > > We will clarify our improved evaluation metric and the rationale behind it in the revised manuscript.

---

> > > > ### Comment · Reviewer_nKX6 · 2025-08-05
> > > >
> > > > Thank you for your response. I acknowledge that the improved TTA demonstrated by the authors proves the effectiveness of their proposed method in addressing low false alarm rates, and as agreed previously, I will raise my final score.
> > > >
> > > > However, I believe the approach of "if the alarm triggers earlier than the 'anomaly appearance', we regard this as a false alarm" is not rigorous. Could this lead to "missed detections"? Only an alarm without an actual incident should be considered a false alarm. If an alarm is triggered before "anomaly appearance" and an incident subsequently occurs, this indicates that the model has long-term prediction capability, which is actually an advantage.
> > > >
> > > > Regarding TTA calculation, I believe the severity of "missed detections" is greater than that of "false alarms." I think the authors should supplement the comparison results of the original TTA, as the original TTA reflects the prediction capability in real application scenarios.
> > > >
> > > > (Due to time constraints, the authors do not need to supplement this experiment during the rebuttal period. The above statements are the reviewer's views. If there are any misunderstandings, the authors can directly correct and refute them, and do not need to fully accept them.)

---

> > ### Author Response · Authors · 2025-08-05
> >
> > Dear Reviewer nKX6,
> >
> > Thank you so much for your recognition of our work and the raised insightful question. We need to clarify that the "anomaly appearance" annotations in the dataset are frames carefully annotated by humans after repeatedly watching the accident videos. These are the earliest moments when a possible accident could be predicted with evidence (i.e., when the collision target first appears in the frame or first exhibits anomalous behavior). It's generally impossible to infer a future accident from frames that occur before this annotation. Therefore, an alert from the model before this point is more likely a **false alarm** (e.g., the target object has not shown up in the camera). To illustrate this point more vividly, we'd like to provide a few case examples:
> >
> > 1. See Figure 2 in the paper. This is an accident where the ego-car collided with a black car that suddenly came out of a blind spot from the left. At 1.6 seconds before the collision, the target black car was not visible in the frame. If the model issued an alert at this time, what is it most likely based on? There are two possible reasons here: (1) The model saw the intersection/cross ahead and guessed that a new car might suddenly appear and collide with our vehicle. (2) The model saw the parked white car next to it and mistakenly thought the ego-car was going to collide with it. Based on our understanding, we lean towards the second reason, where **the target that triggered alerts is not the same as the final collision target**. In this situation, we would consider the alert a false alarm. A similar case can be found in Figure 6 (a) in our supplementary material.
> > 2. For Figure 6 (b), at 2 seconds before the collision, the cyclist who was the final collision target was riding normally along the side of the road and hadn't shown any tendency to cross the street. Same for Figure 6 (c).
> > 3. For Figure 6 (d), at 2 seconds before the collision, the car in front did not brake suddenly, and the ego-car was maintaining a normal following distance. This is a very common scenario in real driving. We do not consider an alert from the model at this time to be an early warning, but rather a false alarm.
> >
> > To further demonstrate that alerts before the "anomaly appearance" are false alarms, we test different methods on **non-accident videos** (normal driving videos with no accident happening). In these cases, any risk alarm by models should be clearly regarded as "false alarm'' without doubt. We show the FAR of these models in the table below (using a recall of 80% on other accident videos as a threshold). This proves that previous methods do indeed have a serious false alarm problem, rather than a genuine "long-term prediction capability".
> >
> > |Method|Drive [1]|CAP [2]|DSTA [3]|GSC [4]|Baseline|Ours|
> > |:-|-|-|-|-|-|-|
> > |FAR|0.59|0.47|0.23|0.19|0.25|0.01|
> >
> > Table 1: False Alarm Rate on Normal Driving Videos (No Accident Occurs)
> >
> > We agree with your statement that "the severity of 'missed detections' is greater than that of 'false alarms'." Indeed, 'missed detections' is measured by the recall rate that we reported in the experiment section (recall rate = 1 - missed detection rate). Therefore, in the table below, we compare the recall rates of different methods on accident videos (using a false alarm rate (FAR) of 0.1 on non-accident videos as the threshold). Our method achieves a 0.93 recall rate (0.07 missed detection rate), while the best prior model only has a recall rate of 0.66 (0.34 missed detection rate). This demonstrates that our method is less prone to missed detections.
> >
> > |Method|Drive [1]|CAP [2]|DSTA [3]|GSC [4]|Baseline|Ours|
> > |:-|-|-|-|-|-|-|
> > |Recall|0.27|0.36|0.61|0.66|0.57|0.93|
> >
> > Table 2: Recall Rate (1-missed detection rate) on Accident Videos
> >
> > Specifically, let's imagine an extrame case where a model consistently outputs a prediction score of 1 (i.e., always predicting risk no matter whatever video frame obseverd), the original TTA metric can reach 5 seconds for a 5-second video data input, but such a model is completely unusable since its false alarm (FAR) is 100%. Therefore, in the paper, we use AUC (the integral of the recall rate at different FARs) as our main evaluation metric, as it provides a more robust assessment of the model's detection capability.
> >
> > In summary, for frames that occur before the "anomaly appearance" annotation, the collision target often hasn't appeared yet or isn't showing any tendency to collide with the ego-car. Therefore, an alert at this time is typically irrelevant to the accident. We will treat these as false alarms. Previous work did not take these real-world situations into account when designing the TTA (Time-to-Accident) calculation metric, which led to these false alarms being misinterpreted as the model's long-term anticipation capability. We believe this is an unreasonable approach that doesn't align with real-world scenarios. We will revise our **manuscript** to make this point clearer.

---

> > > ### Comment · Reviewer_nKX6 · 2025-08-06
> > >
> > > The author's explanation is convincing and has resolved all my questions. The author has also promised to provide open-source code to ensure the reproducibility of the results.
> > >
> > >  I will raise my score from 3 to 5 and suggest that the author incorporate the supplementary information from the rebuttal into the final version.

---

> > > > ### Author Response · Authors · 2025-08-06
> > > >
> > > > Dear Reviewer nKX6,
> > > >
> > > > Thank you for your positive feedback. We are very pleased to hear that our response has addressed your questions and that you find our explanations convincing.
> > > >
> > > > We will definitely incorporate the supplementary information from our rebuttal into the final version of the paper, as you suggested. We also reaffirm our commitment to providing open-source code to ensure the full reproducibility of our results.
> > > >
> > > > We appreciate you taking the time to review our work and provide constructive comments that have helped improve the manuscript.

---

### Official Review · Reviewer_TYgU · 2025-06-25

**Clarity:** 3
**Significance:** 3
**Originality:** 3
**Rating:** 5
**Confidence:** 4

**Summary:**

This study proposes a driving accident anticipation framework that leverages a snippet encoder to capture spatiotemporal dynamics and a Transformer-based decoder to estimate accident probabilities across multiple future time steps. The main contributions are two-fold: (1) the introduction of a refined evaluation protocol that computes recall and Time-to-Accident (TTA) only under acceptable false alarm rates, thereby ensuring practical relevance for real-world deployment; and (2) the development of a predictive framework capable of anticipating the probability of accident occurrence at various future timestamps, offering a more temporally comprehensive risk assessment.

**Questions:**

Please provide a more comprehensive and structured discussion of:
    Works that use TTA or similar temporal safety metrics (e.g., SurvivalNet, DRAG, DAD).
    Studies that model risk as a sequence or through latent continuous variables rather than per-frame labels (e.g., LSTM-based risk predictors, probabilistic risk mapping).

While the paper emphasizes that driving risk "evolves gradually," it’s unclear how this is formalized or measured in your method. What specific representation or signal is used to capture "gradual" risk (e.g., a risk score? probability distribution over time?).

**Ethical Concerns:**

["NO or VERY MINOR ethics concerns only"]

**Limitations:**

Yes

**Quality:**

4

**Strengths And Weaknesses:**

Strengths
Clear and coherent writing: The manuscript is well-organized and accessible to both experts and readers less familiar with the topic.
Relevant and impactful topic: Addressing gradual risk evolution is an important advancement beyond current per-frame models.
Well-structured experiments and results: The use of standard metrics like recall and TTC enables benchmarking, even if their context is underexplored in the related work.

Weaknesses
Incomplete Related Work Section: The manuscript fails to cite or discuss relevant existing works that:
Use recall and Time-To-Accident (TTA) as evaluation metrics in accident anticipation.
Move beyond frame-level analysis by modeling driving risk as a temporally evolving or probabilistic process (e.g., through sequence modeling, latent risk scoring, or uncertainty-aware methods).

This omission makes it difficult to assess the novelty and contribution of the proposed approach. Readers are left without a solid foundation to understand how the presented method compares with or advances the state of the art.

---

> ### Author Rebuttal · Authors · 2025-07-26
>
> Dear Reviewer TYgU,
>
> Thank you for your very positive and constructive review of our submission "Accident Anticipation via Temporal Occurrence Prediction". We greatly appreciate your recognition of our paper's clarity, the relevance and impact of our topic, and the well-structured experiments. Your confidence in our work is truly encouraging.
>
> We acknowledge your valuable feedback regarding the **Related Work section** and the need for a more explicit discussion on how our method formalizes "gradual risk evolution." We agree that addressing these points will significantly strengthen our manuscript, clarify our contributions, and provide a more comprehensive context for readers.
>
> We have carefully considered your questions and would like to address them as follows:
>
> ## **Response to Weaknesses and Questions:**
>
> **1. Comprehensive Discussion of Related Works:**
> Thank you for your valuable comments. In response, we have carefully surveyed and reviewed relevant prior works. We will make sure to include all of them explicitly in the updated manuscript to provide a more comprehensive contextualization of our contribution.
>
> - **Works using recall and Time-To-Accident (TTA) as evaluation metrics:** To the best of our knowledge, all accident anticipation works use recall (AUC or AP) and Time-To-Accident (TTA) (or Time-To-Collision (TTC)) as evaluation metrics. For
> instance, DSA [1] first proposed a dataset and baseline for accident anticipation using AP and TTA. AdaLEA [2] proposed an adaptive loss for early anticipation using AP and TTC. DSTA [3] proposed to utilize dynamic temporal attention and GRU using AP and TTA. DADA [4] proposed to utilize driver attention for anticipation using AUC. CAP [5] proposed a multimodality benchmark for anticipation using AUC, AP, and TTA. Graph(Graph) [6] proposed a nested graph-based framework for early accident anticipation using AP and TTA.
>
> [1] F.-H. Chan et al., “Anticipating accidents in dashcam videos”, in ACCV, 2017.
>
> [2] T. Suzuki et al., “Anticipating traffic accidents with adaptive loss and large-scale incident DB”, in CVPR, 2018.
>
> [3] M. M. Karim et al., “A dynamic spatial-temporal attention network for early anticipation of traffic accidents”, in TITS, 2022.
>
> [4] J. Fang et al., “Dada: Driver attention prediction in driving accident scenarios”, in TITS, 2021.
>
> [5] J. Fang et al., “Cognitive accident prediction in driving scenes: A multimodality benchmark”, in ITSM, 2024.
>
> [6] N. Thakur et al., “Graph(Graph): A nested graph-based framework for early accident anticipation”, in WACV, 2024.
> - **Works modeling driving risk as a temporally evolving or probabilistic process:** UString [7] proposed an uncertainty-based accident anticipation model that employs spatio-temporal relational learning. It sequentially predicts traffic accident probabilities by leveraging Bayesian neural networks to address the intrinsic variability of latent relational representations. DRIVE [8] regarded risk prediction as a Markov decision process and proposed a deep reinforcement learning approach with visual explanation. AccNet [9] formulated the risk by monocular depth-enhanced 3D modeling. CRASH [10] formulated the risk with object detection and context modeling. A branch of work [1, 2, 3, 5, 6, 7, 8] regarded risk evolving as a sequence over time and used RNNs to predict risk. However, the risks before the accident lack accurate labels for training.
>
> [7] W. Bao et al., “Uncertainty-based traffic accident anticipation with spatio-temporal relational learning”, in ACMMM, 2020.
>
> [8] W. Bao et al., “Drive: Deep reinforced accident anticipation with visual explanation”, in ICCV, 2021.
>
> [9] H. Liao et al., “Real-time accident anticipation for autonomous driving through monocular depth-enhanced 3D modeling,” in AA&P, 2024.
>
> [10] H. Liao et al., “CRASH: Crash recognition and anticipation system harnessing with context-aware and temporal focus attentions,” in ACMMM, 2024.
>
> **2. Formalizing and Measuring "Gradual Risk Evolution":** Thank you for the question. We formalize the "gradual evolution" of driving risk as a sequence of future accident probabilities over time. Specifically, for each timestamp, our method predicts a probability sequence at each future timestamp, rather than a single anomaly score. This allows us to model how the likelihood of an accident changes and increases as the vehicle approaches the accident time. We will revise the Method section to clearly explain this issue.
>
> As illustrated in Table 1, the label of the probability sequence evolves from 2.0s before the accident to 0.1s before the accident, like a one-hot vector iteratively shifting left. That's how we formalize the "gradual evolution" of driving risk, while a single anomaly score cannot formalize this evolution.
>
> |Current Timestamp|0.1s|0.2s|0.3s|0.4s|0.5s|0.6s|0.7s|0.8s|0.9s|1.0s|1.1s|1.2s|1.3s|1.4s|1.5s|1.6s|1.7s|1.8s|1.9s|2.0s|
> |:-|:-:|:-:|:-:|:-:|:-:|:-:|:-:|:-:|:-:|:-:|:-:|:-:|:-:|:-:|:-:|:-:|:-:|:-:|:-:|:-:|
> |-3.2s|0|0|0|0|0|0|0|0|0|0|0|0|0|0|0|0|0|0|0|0|
> |-2.0s|0|0|0|0|0|0|0|0|0|0|0|0|0|0|0|0|0|0|0|**1**|
> |-1.4s|0|0|0|0|0|0|0|0|0|0|0|0|0|**1**|0|0|0|0|0|0|
> |-0.7s|0|0|0|0|0|0|**1**|0|0|0|0|0|0|0|0|0|0|0|0|0|
> |-0.1s|**1**|0|0|0|0|0|0|0|0|0|0|0|0|0|0|0|0|0|0|0|
>
> Table 1: Labels of the probability sequence at every timestamp within a future period (0.1s-2.0s), where the accident timestamp is 0.0s.

---

### Official Review · Reviewer_frVz · 2025-07-03

**Clarity:** 3
**Significance:** 2
**Originality:** 3
**Rating:** 4
**Confidence:** 4

**Summary:**

This paper proposes a novel paradigm for driving accident anticipation that predicts the probability of accident occurrence at multiple future timestamps (ranging from 0.1 to 2.0 seconds), rather than estimating frame-level anomaly scores. The technical framework consists of two main components: 1) A snippet encoder to capture spatial-temporal dynamics from past driving frames, and 2) A Transformer-based temporal decoder that estimates accident occurrence probabilities across multiple future steps simultaneously. Additionally, the paper introduces a refined evaluation protocol that measures recall rate and Time-to-Accident (TTA) only under acceptable false alarm rates. Extensive experiments demonstrate the superiority of the proposed method over prior state-of-the-art approaches.

**Questions:**

1. The authors should elaborate on how their approach would perform across diverse environmental conditions and with alternative sensor configurations, potentially providing preliminary results on additional datasets or theoretical justification for expected performance in these scenarios.
2. The baseline comparison is limited to only DRIVE (2021) and CAP (2024), which are not representative of the current state-of-the-art in accident anticipation. The authors should justify this selection and consider expanding comparisons to include more recent and competitive methods to strengthen the evaluation of their approach's relative performance.
3. The authors should provide deeper insights into how far in advance predictions should be made to maximize benefits, and explain the theoretical underpinnings of how predictions at different time steps influence each other, which would strengthen the methodological contribution.
4. The fixed probability threshold approach for warning triggering appears simplistic for real-world deployment. The authors should address how this threshold was determined and discuss the implementation of adaptive thresholding mechanisms that could account for varying driving contexts (urban vs. highway, different times of day, weather conditions).

**Ethical Concerns:**

["NO or VERY MINOR ethics concerns only"]

**Final Justification:**

The authors have addressed most of my concerns in the rebuttal, and their clarifications have improved my understanding of the work. Therefore, I would like to raise my score to 4.

**Limitations:**

Yes

**Paper Formatting Concerns:**

No formatting issues were identified in the paper.

**Quality:**

3

**Strengths And Weaknesses:**

**Strength**：

1. First, the research direction of this paper - Accident Anticipation - is highly significant for autonomous driving with substantial practical application value and social impact.
2. The paper is well-written with clear illustrations and rigorous mathematical formulations to describe the proposed method, making it easy for readers to understand the methodology. Additionally, shifting from anomaly scores to explicit temporal probability predictions for accidents is well-motivated, providing more interpretable and stable supervision.
3. The proposed encoder-decoder framework effectively captures spatio-temporal dynamics, and the proposed evaluation protocol aligns better with real-world deployment needs where excessive false alarms are unacceptable. Furthermore, extensive experiments were conducted on representative datasets, achieving significant performance improvements.

**Weakness**：

1. The experimental section only uses the MM-AU datasets, an ego-view traffic accident dataset. The paper does not discuss how the method might generalize to different environments (e.g., different geographies, lighting/weather conditions, or sensor types).
2. Regarding baseline selection, only DRIVE (2021) and CAP (2024) were chosen, which is too limited, and these baselines are not state-of-the-art anymore.
3. In the method design section, there is insufficient theoretical derivation and analysis of issues such as how far in advance predictions can yield maximum benefits and the dependency relationships between predictions at different time steps. The paper lacks understanding of the mechanisms behind the method.
4. For warning triggering, a fixed probability threshold is used, but there is no explanation for this choice, lacking an adaptive mechanism from a system perspective. Fixed thresholds may not be applicable in different scenarios (e.g., urban/highway, day/night), potentially leading to a surge in false alarms or missed reports in some real-world applications.

---

> ### Author Rebuttal · Authors · 2025-07-25
>
> Dear Reviewer frVz,
>
> Thank you for your thorough review and constructive feedback on our submission "Accident Anticipation via Temporal Occurrence Prediction". We highly appreciate your recognition of the significance of our research direction, the clarity of our paper, and the novelty of our proposed paradigm and evaluation protocol. Your insights are invaluable in helping us improve the quality of our work.
>
> We have carefully considered each of your points and would like to address them as follows:
>
> ## **Response to Weaknesses and Questions:**
> **1. Generalization to Diverse Environments and Sensor Configurations:**
>
> We agree with your valid point that discussing the generalization capabilities of our method is crucial. Here are our points:
> - **MM-AU Includes Diverse Environments:** As stated in [1], MM-AU is a large-scale ego-view traffic accident dataset collected from public accident datasets (CCD, A3D, DoTA, DADA-2000) and various video stream sites (YouTube, Bilibili, Tencent), including diverse weather conditions (sunny, rainy, snowy, foggy), lighting conditions (day, night), scenes (highway, tunnel, mountain, urban, rural), and roadways (arterial, curve, intersection, T-junction, ramp). That's why we chose to train our method on it (unlike previous works primarily use CCD and A3D). Besides, experimental results on the validation subset of MM-AU achieved state-of-the-art, further demonstrating its generalizability across diverse environments.
> - **Nexar Dataset:** We further test our method on the Nexar dataset released by Nexar Inc.. The dataset includes 1,500 annotated video clips of real-world traffic scenarios, encompassing different environmental conditions (lighting, weather) and scene types (urban, rural, highway, etc.). We compare our method with solutions of participants in the Kaggle competition "Nexar Dashcam Collision Prediction Challenge" hosted by Nexar Inc. As shown in Table 2 below, our method outperforms the Champion solution on both public and private leaderboards.
> - **Sensor Types:** Although our current experiments use dashcam data, the snippet encoder could be replaced with other encoders to accommodate input from different sensor modalities (e.g., LiDAR, radar). However, the lack of annotated traffic accident datasets of other sensor modalities prevents us from validating our method's performance with other sensor modalities.
>
> [1] J. Fang et al., “Abductive ego-view accident video understanding for safe driving perception”, in CVPR, 2024.
>
> **2. Baseline Selection and State-of-the-Art Comparison:**
> - **More Baselines:** We further compared our method with two recent SOTA open-source baselines. We trained their model with the same implementation details on MM-AU (CAP-DATA) and demonstrate the experimental results in the table below. The quantitative results in the table demonstrate that our method outperforms recent SOTA methods.
>
> Table 1: Quantitative results comparison of different methods on the CAP-DATA dataset.
> | Method | [C]/[J] | Year | AUC$^{0.1}_{0.0s}$ ↑ | AUC$^{0.1}_{0.5s}$ ↑ | AUC$^{0.1}_{1.0s}$ ↑ | AUC$^{0.1}_{1.5s}$ ↑ | mAUC$^{0.1}$ ↑ | mTTA$^{0.1}$ (s) ↑|
> | :- | :- | :- | :-: | :-: | :-: | :-: | :-: | :-: |
> | DRIVE [2] |ICCV|2021| 0.1288 | 0.1167 | 0.1079 | 0.1231 | 0.1159 | 0.3954 |
> | CAP [3]  |ITSM|2024| 0.0421 | 0.0400 | 0.0296 | 0.0373 | 0.0357 | 0.6372 |
> | DSTA [4]  |TITS|2022| 0.5593 | 0.3862 | 0.2817 | 0.1913 | 0.2864 | 0.8039 |
> | GSC [5]  |TIV|2023| 0.6093 | 0.4177 | 0.2969 | 0.1994 | 0.3046 | 0.8165 |
> | Baseline |-|-| 0.4357 | 0.3938 | 0.2770 | 0.1777 | 0.2829 | 0.5389 |
> | **Ours**  |-|-| **0.8739** | **0.6979** | **0.3892** | **0.2014** | **0.4295** | **0.9327** |
>
> [2] W. Bao et al., “Drive: Deep reinforced accident anticipation with visual explanation”, in ICCV, 2021.
>
> [3] J. Fang et al., “Cognitive accident prediction in driving scenes: A multimodality benchmark”, in ITSM, 2024.
>
> [4] M. M. Karim et al., “A dynamic spatial-temporal attention network for early anticipation of traffic accidents”, in TITS, 2022.
>
> [5] T. Wang et al., “GSC: A graph and spatio-temporal continuity based framework for accident anticipation”, in TIV, 2023.
>
> - **Kaggle Competition (CVPR DriveX Workshop, 2025):** As demonstrated in section B of our supplemental material, we compared our method with other participants' solutions in the **Nexar Dashcam Crash Prediction Challenge [6]**, which is a Kaggle competition of accident anticipation on the Nexar dataset hosted by Nexar Inc., and the Top-3 solutions are announced on the CVPR DriveX Workshop, 2025. As shown in the table below, our method outperforms all the solutions from 282 participants and 237 teams by a large margin, further confirming that our method is state-of-the-art.
>
> Table 2: Public and private scores of our method on the leaderboard of the Nexar Dashcam Crash Prediction Challenge [6].
> | Team | Public Score | Public Rank | Private Score | Private Rank |
> | :- | :-: | :-: | :-: | :-: |
> | MBTDF1 | 0.815 | 4 | 0.788 | 18 |
> | biascia | 0.813 | 5 | 0.826 | 9 |
> | Benchmark | 0.804 | 6 | 0.842 | 5 |
> | Hi F | 0.779 | 13 | 0.853 | 4 |
> | mzwager | 0.853 | 3 | 0.855 | 3 |
> | siuuuuuuuu | 0.886 | 2 | 0.864 | 2 |
> | Paul Endresen | 0.898 | 1 | 0.872 | 1 |
> | **Ours** | **0.924** | - | **0.893** | - |
>
> [6] D. C. Moura et al., “Nexar dashcam collision prediction dataset and challenge”, 2025.
>
> **3. Theoretical Derivation and Analysis:**
>
> Thank you for your valuable suggestions. We will conduct an in-depth analysis of these two points.
> - **How far in advance can predictions yield maximum benefits:**
>
> In our paper, we set the prediction horizon to 2.0 seconds according to our experience. We acknowledge that this selection lacks theoretical analysis. To this end, we conduct comparison experiments of the
> prediction horizon (ranging from 1.0s to 3.0s). As shown in the following table, when the prediction horizon decreases to 1.0s, although the late anticipation performance improves a little, the ability of early anticipation degrades a lot. When the prediction horizon increases to 3.0s, the overall anticipation performance drops. Therefore, we come to the conclusion that setting the prediction horizon to 2.0 seconds yields maximum benefits.
>
> | Prediction Horizon | AUC$^{0.1}_{0.0s}$ ↑ | AUC$^{0.1}_{0.5s}$ ↑ | AUC$^{0.1}_{1.0s}$ ↑ | AUC$^{0.1}_{1.5s}$ ↑ | mAUC$^{0.1}$ ↑ | mTTA$^{0.1}$ (s) ↑|
> | :- | :-: | :-: | :-: | :-: | :-: | :-: |
> |1.0s|**0.8634**|**0.6962**|0.3306|0.0742|0.3670|0.6206|
> |1.5s|0.8496|0.6843|0.3845|0.1564|0.4084|0.7453|
> |2.0s|0.8381|0.6747|**0.3982**|0.2141|**0.4290**|**0.8644**|
> |2.5s|0.8124|0.6685|0.3957|0.2207|0.4283|0.8604|
> |3.0s|0.7953|0.6456|0.3842|**0.2263**|0.4187|0.8532|
>
> - **Dependency relationships between predictions at different time steps:**
>
> Our Transformer-based temporal decoder inherently models the dependencies between predictions at different time steps. The self-attention mechanism in the Transformer allows the model to capture complex, non-linear relationships across the entire predicted sequence. Each output probability is informed by the context of all other positions in the sequence, allowing for a holistic and interdependent prediction. We will revise the Method section of our paper to further elaborate on this dependency.
>
> **4. Fixed Probability Threshold for Warning Triggering:**
>
> You raised a critical point regarding the fixed probability threshold, and we agree that an adaptive mechanism is more aligned with real-world deployment.
> - **Determination of Fixed Threshold:** Following the way of all previous works, the current fixed threshold was empirically determined based on achieving a balance between recall and false alarm rates on the validation set of the MM-AU dataset, aiming for an "acceptable" false alarm rate as per our refined evaluation protocol. This was a pragmatic initial choice for validating our method and a fair comparison with other methods.
> - **Adaptive Thresholding Mechanisms:** We fully concur that a fixed threshold is a simplification for real-world scenarios. We propose to explore adaptive thresholding mechanisms in future work, potentially leveraging:
>     * **Context-aware Thresholds:** Implementing different thresholds based on driving contexts (e.g., urban vs. highway, day vs. night, speed limits). This would require richer contextual information to be fed into the system or a context-aware classification module.
>     * **Uncertainty Estimation:** Integrating uncertainty quantification into our probability predictions. Warnings could then be triggered not just by a high probability, but also by a high probability with low uncertainty, or by dynamically adjusting the threshold based on the predicted uncertainty for a given scenario.

---

> > ### Comment · Reviewer_frVz · 2025-08-07
> >
> > Thanks for the rebuttal provided by the authors. The responses do solve most of my concerns and successfully clarify some details in the paper. I suggest add part of the rebuttal to the revised paper. I would like to raise my score to 4.

---

### Note · Authors · 2025-08-15

Dear Area Chair and Reviewers,

Thank you for your valuable feedback. We have carefully addressed all comments in our rebuttal, which has been acknowledged by the reviewers.

We would like to highlight our key revisions:

- Reviewer frVz: We added new experiments on the Nexar dataset to demonstrate generalizability and provided theoretical justification for our method. The reviewer raised their score to 4.

- Reviewer TYgU: We provided a more comprehensive discussion of related work and clarified how our method formalizes "gradual risk evolution" by predicting a sequence of future probabilities. The reviewer's final rating is 5 (Accept).

- Reviewer nKX6: We clarified the distinction between "score" and "probability" and provided quantitative evidence of False Alarm Rates (FAR) for existing methods. The reviewer's questions were resolved, and they raised their score to 5.

- Reviewer o9V3: We clarified our claims about the novelty of predicting a temporal sequence of risks and provided a detailed comparison to demonstrate that the performance improvement is due to our method. The reviewer acknowledged our responses and is now leaning positive.

We are confident that our rebuttal and the corresponding updates have significantly improved the paper's clarity and robustness. We will incorporate the supplementary information from our rebuttal into the final version of the manuscript.

Thank you again for your constructive feedback.

---

### Decision · Program_Chairs · 2025-09-17

**Decision:**

Accept (poster)

**Comment:**

This work presents an accident anticipation framework that predicts the probability of future accidents at multiple timestamps, instead of assigning frame-level anomaly scores. It integrates a snippet encoder and a Transformer-based temporal decoder to model spatio-temporal dynamics for online prediction. A practical evaluation protocol emphasizes low false alarms while measuring recall and time-to-accident (TTA). Experiments demonstrate strong performance in both recall and TTA, validating the framework’s real-world effectiveness.

The authors’ rebuttal adequately addressed all major concerns raised in the original reviews. Additional experiments were also conducted in response to reviewers’ follow-up questions. These efforts have further strengthened the quality of the paper. Consequently, multiple reviewers raised their scores, leading to a consensus decision to accept the paper. The authors are encouraged to incorporate the new experimental results and discussions into the final version of the paper.